# Promoting equity on licensing exams: Test accommodations for medical students with diabetes

Lucia McGeehan *, Melanie J. Robbins, Daniel Jurich, Sydney Tucker

National Board of Medical Examiners, Philadelphia, Pennsylvania, United States of America

* LMcGeehan@NBME.org

## Abstract

In this article, we will provide information about the USMLE test accommodations process and present data on requests based on diabetes, including the number of requests and the types of accommodations requested and approved. This study summarizes population-level data from 2023 on examinees who requested accommodations on the USMLE due to diabetes, the most prevalent physical impairment cited in requests at the time of this study. The data show that USMLE approves the vast majority of accommodation requests in full and all requests in part from examinees with diabetes. The average processing time is relatively low compared to the communicated 60-business day timeline, with less than 1.1% of cases exceeding this threshold and 42.1% approved in one business day. We conclude by addressing misconceptions that the process for requesting accommodations for individuals with diabetes is arduous and burdensome.

## Introduction

### Overview of USMLE

The United States Medical Licensing Examination (USMLE) is a standardized computer-based examination series comprised of three assessments: Step 1, Step 2 Clinical Knowledge (CK), and Step 3. Successful completion of the USMLE sequence is accepted for licensure of U.S. MD-trained graduates and those completing their training internationally [1,2]. Step 1 assesses the examinee's ability to apply basic science knowledge to the practice of medicine and is a one-day, approximately eight-hour examination [1,2]. Step 2 CK measures an examinee's ability to apply medical knowledge and clinical science essential for supervised patient care and is a one-day, approximately nine-hour examination [1,3]. Step 3 assesses an examinee's ability to apply medical knowledge and an understanding of biomedical and clinical sciences essential for practicing medicine independently and is a two-day examination – day one is approximately seven hours, and day two is approximately nine hours [1,2]. To

**Data availability statement:** The de identified dataset cannot be shared due to ethical and legal restrictions arising from the conditions under which the information was originally collected. As outlined in the USMLE accommodation request form, applicants agree that their submitted information may be used by the USMLE program to evaluate accommodation requests and for research, with the explicit assurance that any information used for research will not include details that could identify them individually and will be reported only in the aggregate. Because participants consented solely to aggregate level reporting, releasing even a de identified dataset would exceed the scope of their agreement. These restrictions reflect both the privacy assurances provided to applicants and the program's legal and ethical obligations related to sensitive examinee data. As a result, the underlying dataset cannot be made available. Data related inquiries may be directed to: library@nbme.org.

**Funding:** The author(s) received no specific funding for this work.

**Competing interests:** The authors have declared that no competing interests exist.

help ensure USMLE results are valid and comparable for all examinees, each Step examination is administered under standard testing conditions [2]. The standardized conditions include the amount of time allotted to complete each assessment section.

## USMLE test accommodations

While the USMLE is administered under standard conditions, there is an exception to this policy to ensure fairness for examinees with documented disabilities who demonstrate a need for accommodations to provide equal access to the examination [3]. The USMLE provides reasonable and appropriate test accommodations to examinees with disabilities, as defined by the ADA [3]. The ADA defines disability as a physical or mental impairment that substantially limits one or more major life activities as compared to most people in the general population [3,4]. Examples of physical or mental impairments include (but are not limited to) epilepsy, diabetes, and visual, hearing, psychiatric, and learning disabilities [3,4]. The major life activities relevant to taking the USMLE include (but are not limited to) seeing, hearing, sitting, concentrating, thinking, reading, and operation of major bodily functions [4].

Requests for test accommodations for the USMLE are individually reviewed and carefully considered by qualified individuals with advanced degrees in psychological and/or medical fields. Appropriate accommodations are provided when examinees demonstrate they are disabled within the meaning of the ADA and need accommodations to take the examination in an accessible manner [3]. Examinees may request accommodations for any Step of the USMLE. Test accommodation decisions are made based on the documentation provided by the examinees. General guidelines and requirements as well as impairment-specific guidelines and requirements detailed on the USMLE website provide transparency around the documentation most relevant to the decision-making process [5]. This guidance includes prompts for drafting personal statements and examples of relevant clinical documentation, including letters from qualified treating professionals detailing the examinee's condition and the specific accommodations recommended. While these recommendations are considered, the final decision is made based on all the information submitted. In some cases, the documentation provided may not fully support the need for the accommodation requested, leading to partial approvals, alternative accommodations being granted, or denial of requested accommodations. Additionally, the USMLE continually solicits and incorporates community feedback on process improvements that streamline the accommodations request process, ensuring a more efficient experience for examinees with disabilities, including those with diabetes.

## Diabetes

Of the approximately 3,500 requests received in 2023 for test accommodations on the USMLE, about 1300 were based on physical impairments. Type 1 and Type 2 diabetes (T1D; T2D) reflected the most prevalent physical impairment cited in requests during this time, with 183 individual requests based on these conditions. T1D, also known as insulin-dependent or juvenile diabetes, is an autoimmune disease characterized by the destruction of insulin-producing beta cells in the pancreas [6]. The most common symptoms of T1D include: frequent urination, increased thirst, increased

appetite with weight loss instead of gain, drowsiness or fatigue, blurred vision, and abdominal pain [6,7]. Type 2 diabetes (T2D), previously known as non-insulin-dependent diabetes mellitus, is a chronic metabolic disorder that occurs when your body does not make enough insulin or does not properly respond to it [8,9]. The most common symptoms of T2D include: frequent urination, increased thirst, increased appetite with weight loss instead of gain, weakness and fatigue, blurred vision, frequent cuts and bruises that are slow to heal, irritability and mood changes, and high levels of sugar in blood when tested [9]. Individuals with T1D and T2D may experience two contrasting blood sugar conditions: hypoglycemia, which refers to blood sugar levels below a healthy range, and hyperglycemia, which indicates high blood sugar [10]. Signs of hypoglycemia include weakness, sweating, chills, rapid heart rate, dizziness, anxiety or irritability, extreme hunger, and trouble concentrating. Symptoms of hyperglycemia include increased thirst, frequent urination, blurred vision, and headache [10]. Maintaining blood glucose levels within a targeted range and regularly monitoring them is crucial for effective diabetes management [11]. Diabetes is a covered disability under the ADA [4]. Therefore, the USMLE provides reasonable and appropriate accommodations to examinees with diabetes who are registered for the USMLE and request accommodations with appropriate supporting documentation.

### Diabetes and USMLE test accommodations

According to the American Diabetes Association, common reasonable accommodations for individuals with diabetes include: breaks (to allow for checking of blood sugar levels, eating snacks, restroom breaks, and taking medication), the ability to keep diabetes supplies nearby, and a private space to test blood glucose or administer insulin [11]. To better serve examinees with diabetes, the USMLE collaborated with their third-party test vendor to expand the items permitted in the secure testing area without the need to request accommodations to include various diabetic supplies [12]. Examinees with diabetes can also request and receive additional break time with standard testing time by submitting a completed and signed request form, with no additional documentation required [13]. Requests for additional break time with standard test time are processed expeditiously within approximately 14 business days (and often sooner than this published timeframe) [13]. In addition to break time with standard testing time, examinees can submit requests for other accommodations, such as break time with shorter testing blocks or additional testing time if necessary for symptom management. Requests other than break time with standard testing time can take approximately 60 business days for review. The purpose of this study is to provide empirical population data related to accommodation approvals and timelines for all examinees requesting USMLE accommodations on the basis of diabetes.

## Methods

### Sample and analysis

Examinees consent to data usage for research purposes when they register for the USMLE as outlined in the USMLE Bulletin of Information and on our website [14]. Data were analyzed from all examinees who submitted accommodations requests for the USMLE Step 1, Step 2 CK, and Step 3 due to diabetes (i.e., diabetes, gestational diabetes, Type 1 diabetes mellitus, Type 2 diabetes mellitus) between January 1, 2023, and December 31, 2023. A total of 183 examinees met this criteria.

We explored three factors related to these requests: the type of accommodation requested, the processing time, and approval outcome. Descriptive statistics were computed to summarize key features of these outcomes, such as the frequency of each type of accommodation requested and granted and the average processing time for requests. All analyses were conducted in Microsoft Excel.

## Results

### Types of accommodations requested

The most commonly requested accommodations were additional break time with standard testing time, additional break time with shorter testing blocks, and additional test time. As shown in Table 1, 116 examinees (63.4%) requested

---

**Table 1. Accommodations Request Outcomes by Request Type.**

| Accommodation Type | Requested | Full Approval | Partial Approval | Denied (%) |
|---|---|---|---|---|
| Additional Break Time | 116 | 116 (100%) | 0 (0%) | 0 (0%) |
| Additional Test Time | 55 | 31 (56%) | 24 (44%) | 0 (0%) |
| Food, Diabetic Supplies, Separate Room | 12 | 12 (100%) | 0 (0%) | 0 (0%) |

additional break time, and 55 examinees (30.1%) requested additional test time. Other requests included approval to bring food, drink, various diabetic supplies, and also to test in a separate room.

## Accommodations granted

Table 1 also displays approval rates for each request. Out of the 183 total requests, 159 examinees (86.9%) received all the accommodations they requested (full approval). Only 24 examinees (13.1%) received different accommodations than requested (partial approval). All 116 examinees who requested additional break time (break time with standard testing time and break time with shorter testing blocks) were approved (full approval: 100%). Of the 55 examinees who requested additional test time, 31 were provided the additional time accommodations they requested (full approval: 56%), 13 were approved for additional time but less than requested (partial approval: 23.6%), and 11 were approved for a form of additional break time in lieu of additional test time (partial approval: 20%). All 12 examinees who only requested food/drink, various diabetic supplies, and a separate room were approved (full approval: 100%).

Full approval rates were high in all Steps, with 74 requests fully approved in Step 1 (78.7%), 52 in Step 2 CK (94.5%), and 33 in Step 3 (97.1%). Partial approvals were less common, with 20 in Step 1 (21.3%), 3 in Step 2 CK (5.5%), and 1 in Step 3 (2.9%). Most partial approvals received break time instead of the additional test time that was requested. No examinee who requested accommodations on the basis of diabetes was denied. The breakdown of full and partial approvals by individual Step exam is provided in Table 2.

## Processing time

Table 3 shows the processing times within and across each Step exam. The processing time for all 183 requests averaged 10 business days. Notably, 77 requests (42.5%) were processed within one day, and 154 requests (84.2%) were processed within 30 business days. All requests for Step 3 (100%), 69 for Step 1 (73.4%), and 50 for Step 2 CK (89.1%) were processed within 30 business days. All requests for Step 1 and Step 3 (100%) and 54 (98.2%) of requests for Step 2 CK were processed within 60 business days. There were two cases which took longer than 60 business days. In total, the average processing times were approximately 13 business days for Step 1, 10 business days for Step 2 CK, and 2 business days for Step 3.

## Discussion

Appropriate accommodations foster a fair and equitable testing environment, where all examinees have the opportunity to demonstrate their knowledge and their abilities without being disadvantaged by their medical condition [15]. The findings

**Table 2. Accommodations Request Outcomes by Step Examination.**

| Exam Step | Total Requests | Full Approval | Partial Approval | Denied |
|---|---|---|---|---|
| Step 1 | 94 | 74 (78.7%) | 20 (21.3%) | 0 (0%) |
| Step 2 CK | 55 | 52 (94.5%) | 3 (5.5%) | 0 (0%) |
| Step 3 | 34 | 33 (97.1%) | 1 (2.9%) | 0 (0%) |
| **Total** | **183** | **159 (86.9%)** | **24 (13.1%)** | **0** (0%) |

**Table 3. Summary of Accommodation Requests and Processing Times.**

| Exam Step | Total Requests | Processed within One Day | Processed within 30 Business Days | Processed within 60 Business Days | Average Processing Time (Business Days) | Median Processing Time (Business Days) | Range (Business Days) |
|---|---|---|---|---|---|---|---|
| Step 1 | 94 | 30 (31.9%) | 69 (73.4%) | 94 (100%) | 13 | 4 | 56 |
| Step 2 CK | 55 | 29 (52.7%) | 51 (92.7%) | 54 (98.2%) | 10 | 1 | 76 |
| Step 3 | 34 | 18 (52.9%) | 34 (100%) | 34 (100%) | 2 | 1 | 17 |
| **Total** | **183** | **77 (42.1%)** | **154 (84.2%)** | **181 (98.9%)** | **10** | **2** | **76** |

from this study provide empirical, population-based data on the USMLE response time and approval rates in addressing accommodation requests for examinees with diabetes. The high approval rate (86.9%) for full accommodations indicates that the majority of examinees were provided with what they requested and deemed reasonable and appropriate, based on their submitted documentation. All examinees received at least a partial accommodation for their exam administration, meaning no examinee who requested accommodations based on diabetes was denied.

The data reveal that additional break time was the most frequently requested and fully approved accommodation, with all 116 examinees receiving the requested break time (100% full approval). This reflects the specific needs of examinees with diabetes to manage their condition during the lengthy examination period and is consistent with the most common reasonable accommodation recommended by the American Diabetes Association [11]. The ability to take additional breaks is crucial for these examinees to monitor their blood glucose levels, administer insulin, and consume food or drink as needed to maintain their health and performance during the exam.

The approval rate for additional test time, although lower in full approval (56%), still demonstrates a significant effort to accommodate the needs of these examinees. The partial approvals for additional test time (23.6% for less time than requested and 20% for additional break time instead) suggest a tailored approach to meet individual needs while maintaining the fairness and integrity of the examination process. Additional time was generally denied in cases where the request was based on an additional impairment that was not supported by the documentation, or where there was insufficient rationale from the treating professional and/or the examinee explaining why extended testing time, rather than additional break time, was necessary to manage the reported symptoms. These results show that while some requests for additional test time were not fully granted, alternative accommodations were provided so that examinees could still manage their diabetes effectively.

The processing time, averaging 10 business days, with 42.5% of requests processed within one day, highlights the commitment of the USMLE to support examinees in a timely manner. The prompt processing of requests ensures that examinees have sufficient time to prepare for the exam with the knowledge that their accommodation needs will be met.

## Limitations

This study has some limitations that should be considered when interpreting the results. First, the study focuses solely on examinees with diabetes. Findings should not be assumed to generalize to other conditions. Second, the data were obtained from a single year (2023), and trends may vary over time. Third, the study relies on the accuracy and completeness of the documentation provided by examinees and their treating professionals, which may affect the decision-making process. Finally, the study does not explore the experiences of examinees who chose not to apply for accommodations, which could provide valuable insights into areas for improvement.

## Conclusions

The USMLE remains committed to providing reasonable and appropriate accommodations to the diverse population of test takers, enabling equal and fair examination access. This study shares data on the USMLE accommodation request

outcomes for examinees with diabetes. The data show that USMLE approves a high majority of requests in full and all requests in part for accommodations from examinees with diabetes. The high approval rates for additional break time and the tailored approach to additional test time requests demonstrate a nuanced approach to the specific needs of examinees with diabetes. The average processing time is relatively low compared to the communicated 60-business day timeline, with less than 1.1% of cases exceeding this threshold. The USMLE strives to provide clear communication and updated information on the request process for individuals in the medical educational space, and is actively working to streamline various processes to further accelerate responsiveness, ensuring all examinees with documented disabilities receive timely and appropriate accommodations.

## Supporting information

**S1 File. NBME Institutional Research Plan Blanket IRB Determination.**
(PDF)

## Author contributions

**Data curation:** Melanie J. Robbins.

**Methodology:** Melanie J. Robbins.

**Project administration:** Lucia McGeehan.

**Supervision:** Lucia McGeehan.

**Writing – original draft:** Lucia McGeehan, Melanie J. Robbins, Daniel Jurich.

**Writing – review & editing:** Sydney Tucker, Daniel Jurich.

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
