## [Decision Letter · Decision Letter 0]

30 Dec 2025

Promoting equity on licensing exams: Test accommodations for medical students with diabetes

PONE-D-25-24269

Dear Dr. McGeehan,

We’re pleased to inform you that your manuscript has been judged scientifically suitable for publication and will be formally accepted for publication once it meets all outstanding technical requirements.

Kind regards,

Fateen Ata, MD

Academic Editor

PLOS One

**Journal requirements:**

Reviewers' comments:

Reviewer's Responses to Questions

**Comments to the Author**

1. Is the manuscript technically sound, and do the data support the conclusions?

Reviewer #1: Yes

Reviewer #2: Yes

Reviewer #3: Yes

2. Has the statistical analysis been performed appropriately and rigorously?

Reviewer #1: Yes

Reviewer #2: Yes

Reviewer #3: Yes

3. Have the authors made all data underlying the findings in their manuscript fully available?

Reviewer #1: Yes

Reviewer #2: Yes

Reviewer #3: Yes

4. Is the manuscript presented in an intelligible fashion and written in standard English?

Reviewer #1: Yes

Reviewer #2: Yes

Reviewer #3: Yes

Reviewer #1: The Article in terms of its set out aims, and corresponding objectives has brought out the inferences and conclusions commensurate with the set out aims and objectives with matcheable standard methodology. The conclusions / inferences are loud and clear specially bringing out that the USMLE as an examining statutory authority in terms of its policy with reference to accommodation deals with the problem of diabetic examinee accommodation appropriately and more importantly, timely, which are the key cornerstones on this count. The answers that have been put across to the posers are based on sound and critical appraisal of the article on merit, sumptuousness, clarity and being in tune with the governing principles of research methodology. The limitations pertaining to the study need to be tided over in future research studies by incorporating leftover confounding variables in the present study. As such, the study merits publication in my opinion and depicted review.

Reviewer #2: The paper involved diabetes-relaed testing accomodations. The writing is good, the data are sound, and the conclusions are reasoable. The references seem sufficiently comprehensive. Accept. The paper is publishable, the methodology and writing are sound.

Reviewer #3: The manuscript on diabetes accommodation is an important and timely contribution, as it demonstrates how thoughtful, evidence-based accommodations can significantly support candidates with diabetes in successfully completing state examinations, such as the USMLE.

This should not be viewed as a weakness. The USMLE administration recognizes that certain medical conditions can affect performance during state examinations, and therefore considers appropriate accommodations to ensure fair and equitable assessment.

**Do you want your identity to be public for this peer review?** For information about this choice, including consent withdrawal, please see our Privacy Policy

Reviewer #1: **Yes:** Dr. Vedprakash Mishra Honorary Chief Advisor Datta Meghe Institute of Higher Education and Research (Deemed to be University), Nagpur, Maharashtra, India.

Reviewer #2: No

Reviewer #3: No

---

## [Editor Report · Acceptance letter]

PONE-D-25-24269

PLOS One

Dear Dr. McGeehan,

I'm pleased to inform you that your manuscript has been deemed suitable for publication in PLOS One. Congratulations! Your manuscript is now being handed over to our production team.

Kind regards,

on behalf of

Dr. Fateen Ata

Academic Editor

PLOS One